# Effect of Seminal Plasma on the Freezability of Boar Sperm

**DOI:** 10.3390/ani14243656

**Published:** 2024-12-18

**Authors:** Kuanfeng Zhu, Yukun Song, Zhi He, Peng Wang, Xuguang Wang, Guoshi Liu

**Affiliations:** 1College of Animal Science, Xinjiang Agricultural University, Urumqi 830091, China; zhufengzi123@dingtalk.com (K.Z.); hezhi626@126.com (Z.H.); 2Beijing Jingwa Agricultural Science & Technology Innovation Center, Beijing 101205, China; songyukun@cau.edu.cn (Y.S.); wp507810@cau.edu.cn (P.W.); 3National Engineering Laboratory for Animal Breeding, Key Laboratory of Animal Genetics and Breeding of the Ministry of Agriculture, Beijing Key Laboratory for Animal Genetic Improvement, College of Animal Science and Technology, China Agricultural University, Beijing 100193, China

**Keywords:** seminal plasma, boar, proteome, metabolome, freezability

## Abstract

Seminal plasma (SP) is a crucial component of semen that significantly influences sperm function. However, the relationship between SP and the freezability of boar sperm remains underexplored compared to the functional diversity of SP. Utilizing metabolomics and proteomics approaches, we identified 13 differentially expressed metabolites, predominantly lipids such as phosphoethanolamine, and 38 proteins including CRYAA, CUTC, SHANK1, PFN1, NEU1, SAA3, TACSTD2, APOA2, and CCN6. These metabolites and proteins were enriched in processes and functions related to cytoskeleton dynamics and cell adhesion. Notably, 33 metabolites showed significant correlations with the average progressive motility (PM) at 10 min and 2 h post-thawing. Among these, seven negatively correlated metabolites, including myrisamine oxide and minoxidil, were identified as drugs or environmental pollutants, while positively correlated metabolites included glycerol phosphocholine, creatine, and carnitine. Furthermore, we identified 70 related proteins enriched in gene ontology (GO) terms associated with cell division and cycle regulation, as well as KEGG pathways involving thermogenesis and pyruvate metabolism. The metabolites and proteins linked to average PM at 10 min and 2 h post-thawing were jointly enriched in pathways related to thermogenesis, arginine and proline metabolism, and ether lipid metabolism. Additionally, six reproductive hormones, including testosterone and estradiol, were detected in SP using ELISA; however, none showed significant correlations with semen quality before or after freezing. When using highly and lowly freezable SP as base freezing extenders, the protective effect of highly freezable SP was not significantly superior to that of lowly freezable SP, and it did not outperform the control group using a commercial freezing extender.

## 1. Introduction

Seminal plasma (SP) is the liquid component of semen, excluding sperm, and constitutes the primary component by weight in the semen of various species, including pigs, horses, and dogs. It is composed of nutrients, salts, and a diverse array of proteins, nucleic acids, and lipids, all of which significantly influence sperm motility, energy acquisition, and various reproductive functions, including fertilization. Notably, SP contains epididymal fluid and secretions from accessory reproductive glands, such as the prostate and bulbourethral glands. In boars, the majority of SP is derived from the prostate. The epididymis serves as the site for sperm maturation, and the epididymal fluid plays a crucial role in the functional maturation of sperm. Prostaglandins present in prostatic fluid can enhance the activity of both sperm and cells within the female reproductive tract. Additionally, the components of SP can interact with sperm through mechanisms such as exocytosis, diffusion, and fusion, further modulating sperm function. Given its diverse effects on sperm, SP may also influence sperm freezability.

The metabolites of Huoshou Black boar seminal plasma (SP) [1] have been linked to sperm freezability. Similarly, the fructose and other components found in rooster SP [2] have shown correlations with sperm freezability. Xu Bingbing altered the freezing outcomes by exchanging SP from highly and lowly freezable Inner Mongolia cashmere goats [3]. However, some studies have indicated that boar SP possesses a certain degree of autologous toxicity that may affect semen preservation [4]. It remains to be directly verified whether boar SP can provide a protective role during the freezing process. Therefore, this experiment is designed to investigate the relationship between SP and sperm freezability through proteomic and metabolomic analyses. The protective effect of SP on boar sperm cryopreservation will be directly validated using highly freezable and lowly freezable SP as cryoprotective solutions.

## 2. Materials and Methods

### 2.1. Equipment and Reagents

The semen was obtained from a pig farm in Pinggu, Beijing, where the boars were of the French Landrace and Yorkshire breeds. The experimental period spanned from May 2023 to May 2024.

The CASA system (TYCASA.V1.0), programmed balancer (TYPHY10), automatic sealing and filling machine (TYGZJ-1-50), programmed freezer (TYLDY-900), large-capacity centrifuge (TYLXJ-0-4500), 4 °C operation cabinet (TYDWCZG-150), and 100 L self-pressurized liquid nitrogen tank (BGHN-ZZYG-100) were all supplied by Beijing Tianyuanaorui Biotechnology Co., Ltd., (Beijing, China). Additionally, a 35 L liquid nitrogen tank was provided by Jinfeng. The ST-360 Microplate System was sourced from Shanghai Kehua Biological Engineering Co., Ltd., (Shanghai, China)., along with the Protein horizontal and transfer electrophoresis tank (Mini-TB4), electrophoresis apparatus (DYY-8C), and chemiluminometer (Tanon 5200).

The boar semen freezing reagent kit (including boar semen freezing reagents ABCDE) was supplied by Beijing Tianyuanaorui Biotechnology Co., Ltd. Other materials included ultra-pure water (Merck Millipore, Shanghai, China), a testosterone/progesterone/prohormone/estradiol/prolactin/prostaglandin ELISA kit (48T, Beijing Jinhaikeyu Biology, Beijing, China), acetonitrile (ThermoFisher Scientific, Shanghai, China), and formic acid (Fluka, Bryanz, Switzerland). Proteinome-related reagents are detailed in Appendix A.

Ancillary equipment included a 0.50 mL straw (Minitube), 500 mL tip bottom centrifugal bottle (Corning), 4 mL EP centrifuge tube, and 10 μm ruby counting plate (Shanghai Weitu).

### 2.2. Methods

#### 2.2.1. Solution Preparation

Boar semen freezing extenders were prepared according to the instructions of the supplier. Pre-diluent: One package of boar semen freezing reagent A was dissolved in 1 L of ultra-pure water and prepared before semen collection, then pre-warmed at 37 °C for use. Freezing extender I: One bottle of boar semen freezing reagent B (mainly contains lactose and trehalose) was dissolved in 400 mL of ultra-pure water, mixed with 100 g of egg yolk, and then centrifuged at 4000× *g* for 20 min. The supernatant was collected, refrigerated at 4 °C, and equilibrated to 17 °C before use. Freezing extender II: 276 mL of freezing extender I was taken, and 18 mL of boar semen freezing reagent C (glycerin) along with 4.2 mL of reagent D were added. The mixture was fully combined and divided into 80 mL insemination bottles, then stored at 4 °C. Thawing solution: a bottle of reagent E was dissolved in 1 L ultra-pure water and then refrigerated at 4 °C.

In the experiment employing SP as base extender, freezing extender I was formulated using 80% SP (either high or low freezability) and 20% egg yolk. Freezing extender II was then prepared using freezing extender I for the treatment groups (H or L).

#### 2.2.2. Semen Freezing and Thawing

Semen was collected by hand methods and immediately diluted with 37 °C pre-diluent at a ratio of 1:2 by volume. A total of 3 to 6 ejaculations were collected and placed in a foam box with ice packs for transport to the laboratory, which took approximately 40 min. Upon arrival, 1 mL of semen was extracted to assess the original concentration, progressive motility (PM), and total motility (TM). Semen samples with a concentration greater than 2 × 10^8^ and an abnormal rate of less than 10% were selected for the experiment and equilibrated at 17 °C for 1 h. The samples were then centrifuged at 17 °C and 850× *g* for 13 min. The supernatant was discarded, and the pellet was diluted to 2 billion/mL with equilibrated freezing extender I at 17 °C. The samples were transferred to a programmed balancer, where the cooling curve was set to decrease from 17 °C to 4 °C over 90 min. Once cooled to 4 °C, the semen was moved to a 4 °C operation cabinet, diluted 1:1 with pre-cooled freezing extender II at 4 °C, and gently mixed. The mixture was then filled using an automatic filling and sealing machine and labeled with a marker. The freezing curve was as follows: cool from 4 °C to 1 °C over 2 min; then from 1 °C to −140 °C over 4.7 min; hold at −140 °C for 5 min. Finally, the samples were stored in a liquid nitrogen tank.

The semen was thawed at 60 °C for 15 s in a water bath and then diluted with 7 times its volume of pre-warmed thawing solution at 37 °C. Motility and movement parameters were analyzed after 10 min and 2 h of incubation at 37 °C.

#### 2.2.3. Semen Analysis

The temperature of the microscope objective table was set at 37–38 °C, and the ruby counting board was preheated on the objective table in advance. The samples to be inspected were preheated to 37 °C for 10 min, inverted approximately 30 times for thorough mixing, and then quickly placed as a 10 μL drop on the ruby counting plate for about 10 s. Four fields from different positions were selected under the 10× negative phase objective, and the average values were calculated. Progress motility (PM), total motility (TM), and movement parameters (including VCL: velocity of the curvilinear path; VSL: velocity of straight line; VAP: velocity of the average path; ALH: amplitude of lateral head; STR: straightness; WOB: wobble; LIN: linearity) were assessed.

CASA system parameter settings: frame interval 20 ms, exposure time 10 ms, 25 frames; TM threshold VCL > 35 μm/s; PM threshold “VSL > 25 μm/s” or “10 μm/s < VSL < 25 μm/s and LIN > 0.4”.

#### 2.2.4. Grouping of Boars and Separation of SP

At the beginning of the experiments, semen from approximately 30 boars was frozen 2 to 4 times for each boar over a period of about half a month. The boars were then sorted based on the mean progress motility (PM) measured at 10 min and 2 h after thawing during each semen collection cycle. A boar was classified into the high or low freezability group if it ranked in the top or bottom 50% at least twice. If a boar ranked in the top twice and in the bottom twice, it was not classified into either group. Ultimately, 11 boars were selected as high freezability and another 11 as low freezability. Concurrently, all the seminal plasma (SP) was separated by centrifugation at 1000 rpm for 13 min, then filtered through 3 μm, 1 μm, and 0.22 μm PVDF filters, and frozen at −80 °C for later use. Thus, when the boars were selected, the corresponding SP was also selected.

#### 2.2.5. Detection of Seminal Plasma Reproductive Hormone

Six reproductive hormones, namely, testosterone (T), estradiol (E2), luteinizing hormone (LH), progesterone (P), prolactin (PRL), and prostaglandin F2α (PGF2α), were tested using the ELISA method, following the kit instructions. In the Excel worksheet, the standard values and their corresponding optical density (OD) values were used to create scatter plots. A linear regression curve was then calculated, allowing for the determination of each sample’s concentration based on its OD value.

#### 2.2.6. Protein Extraction, Digestion, and LC-MS/MS Analysis

Each sample was prepared by mixing the seminal plasma from three boars in equal volumes for both proteomic and metabolomic analyses.

Protein Extraction: Samples were removed from −80 °C and centrifuged at 4 °C at 12,000× *g* for 10 min. The supernatant was transferred to a new centrifuge tube, and protein concentration was determined using a BCA kit. Protein samples were then processed using SDS-PAGE gel electrophoresis at 170 V for 55 min, with 15 μL of sample loaded into each well. The gels were stained with Coomassie blue to preliminarily assess sample quality.

Pancreas Lysis: Equal amounts of protein from each sample were taken, and the volume was adjusted to match that of the lysate. One volume of precooled acetone was added, followed by vortexing. Subsequently, four volumes of precooled acetone were added, and the mixture was precipitated at −20 °C for 2 h. Afterward, the samples were centrifuged at 4500× *g* for 5 min, and the supernatant was discarded. The precipitate was washed 2–3 times with precooled acetone at −20 °C. After drying the precipitate, TEAB was added to achieve a final concentration of 200 mM. The precipitate was then dispersed using an ultrasonic oscillator, and trypsin was added at a ratio of 1:50 (protease: protein, m/m) for overnight digestion. Dithiothreitol (DTT) was added to a final concentration of 5 mM and reduced at 56 °C for 30 min. Finally, iodoacetamide (IAA) was added to a final concentration of 11 mM and incubated at room temperature for 15 min.

LC-MS/SMS Analysis: Peptides were resolved by liquid chromatography using an Easy-nLC1000 ultra-efficient liquid chromatography system. Mobile phase A consisted of an aqueous solution containing 0.1% formic acid and 2% acetonitrile, while mobile phase B was an acetonitrile–aqueous solution containing 0.1% formic acid. The liquid phase gradient was set as follows: 0–14 min, 6–24% B; 14–16 min, 24–35% B; 16–18 min, 35–90% B; 18–20 min, 90% B, with a constant flow rate of 500 nL/min. The peptides were separated through the ultra-efficient liquid chromatography system and injected into the capillary ion source for ionization, followed by data acquisition in the timsTOF Pro mass spectrometer. The ion source voltage was set to 1.75 kV, and both the peptide parent ions and their secondary fragments were detected and analyzed using Time-of-Flight (TOF) mass spectrometry. The data acquisition mode utilized data-independent acquisition with parallel accumulation serial fragmentation (dia-PASEF). The primary mass spectrum scan range was set to 300–1500 *m*/*z*, with 20 PASEF acquisitions, and the secondary mass spectrum scan was conducted within a 400–850 *m*/*z* interval, using a 7 *m*/*z* window. DIA data were processed using the DIA-NN search engine (v.1.8). Tandem mass spectra were searched against the Sus_scrofa_9823_PR_20240407.fasta database (46,176 entries), concatenated with a reverse decoy database. Trypsin/P was specified as the cleavage enzyme, allowing for up to one missed cleavage. N-terminal methionine excision and carbamidomethylation of cysteine were specified as fixed modifications, with a false discovery rate (FDR) adjusted to <1%.

#### 2.2.7. Metabolite Extraction and LC-MS/MS Analysis

Metabolite Extraction: Samples were removed from −80 °C and slowly thawed. Four volumes of extraction buffer (MeOH/ACN, 1:1, *v*/*v*) were added, followed by sonication. After precipitating at −20 °C for 1 h, the samples were centrifuged at 18,000× *g* at 4 °C for 15 min to remove protein precipitates. The supernatant was then transferred to a new centrifuge tube, and a quarter volume of ACN: H_2_O (1:1, *v*/*v*) was added. The mixture was centrifuged again at 18,000× *g* at 4 °C for 15 min, and the supernatant was transferred to a new centrifuge tube for LC/MS analysis.

LC-MS/MS Analysis: Metabolites were separated using a Waters UPLC ultra-efficient liquid chromatography system, combined with a Waters ACQUITY UPLC BEH C18 column (1.7 µm, 2.1 mm × 100 mm). A sample volume of 10 µL was injected, and metabolites were eluted at a flow rate of 400 µL/min with a column temperature set to 45 °C. Mobile phase A consisted of an aqueous solution containing 0.1% formic acid, while mobile phase B was acetonitrile containing 0.1% formic acid. The liquid phase gradient was set as follows: 0–11 min, 2–98% B; 11.0–12.0 min, 98% B; 12.0–12.1 min, 98–2% B; and 12.1–15.0 min, 2% B. After separation, metabolites were injected into the ESI ion source for ionization and subsequently analyzed using the timsTOF Pro mass spectrometer. The ion source voltage was set to 4.5 kV, allowing for the detection and analysis of both the parent ions and their secondary fragments using high-resolution Time-of-Flight (TOF) mass spectrometry. The mass spectrometry scan range was set to 50–1300 *m*/*z*. Data acquisition was performed in parallel cumulative serial fragmentation (PASEF) mode, with a first-order mass spectrum collected in the range of 0–1 in PASEF mode. The dynamic exclusion time for tandem mass spectrum scanning was set to 6 s to prevent repeated scanning of the same parent ion.

Database Retrieval: Mass spectrometry data were processed using MetaboScape 2022 for peak extraction, alignment, and retention time correction of the raw data. Primary and secondary quality errors were maintained within 20 ppm to ensure the accuracy of the identification results. The structure and annotation information of metabolites were obtained through comparisons with the NIST, HMDB, and our database, as well as spectral map comparisons from integrated public databases.

#### 2.2.8. Bioinformatics Methods

Based on the quantitative information of metabolites and proteins obtained from database matching, we calculated the metabolite fold changes between the high (H) and low (L) groups. VIP (Variable Importance in Projection) values were calculated using the *p*-values from univariate *t*-test analysis, along with multivariate statistical analysis and orthogonal partial least squares discriminant analysis (OPLS-DA). This analysis led to the identification of significantly differentially expressed metabolites (DEMs) and proteins (DEPs). Subsequently, we performed expression clustering, Gene Ontology (GO) analysis, and KEGG enrichment analysis for the identified DEMs and DEPs.

#### 2.2.9. Statistically Analysis

Statistical analyses were conducted using SPSS 23.0. Sperm motility and movement parameters were analyzed using ANOVA followed by Dunnett’s test, with a *p*-value of 0.05 considered statistically significant. For protein expression analysis, mean data were used to calculate the fold change. Proteins with fold changes >1.5 or <1/1.5, along with *t*-test *p*-values < 0.05, were deemed significantly regulated. Correlation coefficients were calculated using Pearson correlation. Data for sperm motility and movement parameters are presented as “mean ± SEM”. The notations “ns”, “,” “,” and “ represent “no significant”, “*p* < 0.05”, “*p* < 0.01”, and “*p* < 0.001” compared to the control, respectively.

## 3. Results

### 3.1. Comparison of the Motility of Highly and Lowly Freezable Group Before and After Freezing

For the average results before semen freezing, there was no significant difference in PM (52.42% ± 3.6% vs. 52.4%) and TM (85.31% ± 2.1% vs. 80.37% ± 3.17%) between the high (H) and low (L) freezability groups (*p* > 0.05). However, for the average results 10 min and 2 h after semen thawing, the PM of the H group was significantly higher (*p* < 0.01) than that of the L group (49.81% ± 1.04% vs. 39.57% ± 1.62%), while the TM showed an extremely significant difference (*p* < 0.001) (59.03% ± 1.3% vs. 47.53% ± 2.16%) (Figure 1).

### 3.2. Correlation Between Reproductive Hormone in SP and Motility Before and After Freezing

As shown in Table 1, there was no significant (*p* > 0.05) correlation between reproductive hormones (T, P, LH, PRL, E2, PGF2α, T/E2, P/E2) and semen quality indexes (including TM, PM after predilution, original concentration, TM, PM at 10 min and 2 h after thawing).

### 3.3. Metabolome Analysis of Highly and Lowly Freezable Semen Plasma

The quality control group (QC) exhibited a high degree of concentration, indicating an effective detection procedure (Figure 2A). However, the sample distribution was relatively dispersed. Principal component analysis (PCA) revealed that Group H was not completely separated from Group L (Figure 2C). The correlation coefficient between the samples ranged from 0.865 to 0.964 (Figure 2D), and the correlation between the two groups was not greater than that between the groups. In the relative standard deviation (RSD) diagram (Figure 2B), the mean coefficient of variation for both groups was greater than 0.2, with values concentrated between 0.15 and 0.33 for Group H and between 0.15 and 0.35 for Group L.

A total of 348 metabolites were identified, primarily consisting of hormones and transmitters. The metabolites with the highest concentrations included glycerophosphate choline, carnitine and its derivatives, and oligopeptides (gln-lys-leu). At *p* < 0.05 and fold_change > 1.5, 12 differentially expressed metabolites (DEMs) were selected (Table 2), of which 9 were upregulated and 3 were downregulated in Group H compared to Group L. The DEMs were predominantly lipids, including four phosphoethanolamines (PE), two phosphorylcholines, two sphingomyelins (SM), and one phosphatidylserine (PS).

In total, 33 metabolites were associated with the mean PM (MAMPs) at 10 min and 2 h after thawing, representing 9.5% of the total identified metabolites. Among these, 7 were negatively correlated and 26 were positively correlated (Appendix A). The metabolites that positively correlated with thawing PM included six phospholipids (three PEs, with PE 21:4 nearly significant; and three phosphorylcholines), four oligopeptides, three sphingolipids, phosphatidylserine (PS 34:2), five organic acids or acid anhydrides (including creatine and S-adenosylhomocysteine), and seven others. The metabolites that were negatively correlated included Myristamine Oxide, Minoxidil, C17-Sphinganine, Octenoyl-carnitine, 2-Hydroxy-4-(octyloxyl)benzophenone, Spisulosine, and 2-ethylsulfanyl-N-[[1-(hydroxymethyl)cyclopropyl]methyl]benzamide.

### 3.4. Proteome Analysis of Highly and Lowly Freezable Semen Plasma

The signal intensity was generally strong, with a ratio between 4.6% and 5.6% below 10^2.8^ (Figure 3A). The 4129 detected peptides were concentrated at 7 to 21 amino acids (Figure 3B), demonstrating the highest accuracy, which indicates that the detection results were reliable. The average relative standard deviation (RSD) for group H was 0.19, with a concentration range of 0.11 to 0.31, while for group L, it was 0.23, with a concentration range of 0.14 to 0.35. The PCA plot indicated that the H and L groups were not significantly separated, and the heatmap showed that the correlation between samples in both groups was above 0.94. A total of 1000 proteins were identified, of which 980 were comparable.

A total of 38 differentially expressed proteins (DEPs) were identified with a significance level of *p* < 0.05 and FC > 1.5, accounting for 3.8% of the total proteins identified. Among these, 20 proteins in group H were highly expressed compared to group L, while 18 were lowly expressed (Appendix A). The top 10 highly expressed proteins with the largest fold changes were CRYAA, CUTC, SHANK1, PFN1, ENPP2, PLS1, CRYBB2, CYLC2, HIP1, and PLA1A. Conversely, the top 10 lowly expressed proteins were CCN6, APOA2, TACSTD2, NEU1, ADIRF, SBSN, EDA, ZG16B, GALNT18, and VTN.

The top 20 gene ontology (GO) items with significant enrichment of DEPs comprised 6 biological processes (BP), 6 cellular components (CC), and 8 molecular functions (MF). The most significant BP entries (Figure 4D) included GO:0032232, which pertains to the negative regulation of actin filament bundle assembly, and GO:0051497, related to the negative regulation of stress fiber assembly, among others. The most prominent CC items (Figure 4E) were associated with the cytoskeleton and cell junctions, including GO:0005923 for bicellular tight junction, GO:0070160 for tight junction; GO:0030864 for cortical actin cytoskeleton and GO:0030863 for cortical cytoskeleton. The most prominent MF items (Figure 4F) were associated with structure and cadherin, including GO:0005198 for structural molecule activity; GO:0098641 for cadherin binding involved in cell-cell adhesion; GO:0098632 for cell-cell adhesion mediator activity; GO:0098631 for cell adhesion mediator activity; GO:0005200 for structural constituent of cytoskeleton; and GO:0045296 for cadherin binding. Additionally, only one KEGG pathway (Figure 4G) was enriched: the cytokine–cytokine receptor interaction, specifically the interaction between EDA and its receptors EDAR and XEDAR within the TNF family pathway.

In total, 70 proteins were associated with the mean PM (PAMPs) at 10 min and 2 h after thawing. Among these, 50 proteins were positively correlated, while 20 were negatively correlated. Notably, 29 proteins exhibited significant differences between groups H and L (Appendix A), with 17 showing positive correlations and 12 showing negative correlations. Additionally, 19 proteins had a fold change (FC) > 1.5 or <1/1.5, comprising 9 positive correlations and 10 negative correlations. Furthermore, 41 proteins were significantly correlated but did not show significant differences (Appendix A).

GO_BP enrichment analysis revealed numerous connections between PAMPs and processes such as cell division and cycle regulation (Figure 5A–D). Additionally, three KEGG pathways were enriched (Figure 5E), namely, map00620 (Pyruvate metabolism), map04380 (Osteoclast differentiation), and map04714 (Thermogenesis).

Proteins and metabolites associated with the mean PM at 10 min and 2 h after thawing were jointly enriched in three KEGG pathways (Figure 6): ssc00330 (Arginine and proline metabolism), which involved one protein, acetaldehyde dehydrogenase family 9 A1 (ALDH9A1), and one metabolite, creatine; ssc04714 (Thermogenesis), which included two proteins, cAMP response element binding protein 3 (CREB3) and actin β/γ1 (ACTB_G1), along with one metabolite, carnitine; and ssc00565 (Ether lipid metabolism), which involved one protein, phosphodiesterase family 2 (ENPP2), and one metabolite, glycerophosphocholine.

### 3.5. Effect of SP as Base Freezing Extender on Thawing Motility and Movement Parameters of Frozen Semen

At 10 min post-thawing, there were no significant differences in total motility (TM), progressive motility (PM), the velocity of curve line (VCL), the velocity of straight line (VSL), velocity of path rate (VAP), amplitude of lateral head (ALH), straightness (STR), and wobble (WOB) between the treatment groups and the control. However, the H group showed a slight improvement over the L group, although this difference was not statistically significant. For linearity (LIN), the L group exhibited a significantly lower value compared to the control, and it was also lower than the H group, although this difference was not significant. The difference between the H group and the control was not significant.

At 2 h post-thawing, total motility (TM) and velocity of the curve line (VCL) in the L group were significantly lower than in the control group (*p* < 0.05). Additionally, PM, VAP, VSL, STR, WOB, and LIN in the L group were all significantly lower than in the control group (*p* < 0.01, 0.001, or 0.0001). In the H group, PM and STR were lower than in the control group (*p* < 0.05), while VSL, WOB, and LIN were significantly lower (*p* < 0.01 or 0.001) compared to the control. Overall, the H group showed slightly higher values than the L group across all indicators, but these differences were not statistically significant.

## 4. Discussion

Since semen freezing results can fluctuate with each batch, 30 boars were tested at least twice during the same period. Boars that exhibited a relatively stable mean PM at both 10 min and 2 h post-thawing were selected. It is important to note that the screening of boars with highly and lowly freezable sperm was not based on the arithmetic mean of freezing results. As a result, the average thawing PM and TM of some boars in the H group were lower than those in the L group. However, the thawing PM and TM of most boars in the H group were higher than those in the L group, indicating that the screening method was effective.

Reproductive hormones, including E2, T, PRL, P, PGF2α, and LH, play important roles in spermatogenesis, sperm maturation, and the fertilization process. However, no significant correlations were detected between these hormones and semen quality indicators, including pre-dilution PM and TM, original concentration, and PM and TM at both 10 min and 2 h post-thawing. This suggests that these hormones may not have a simple linear association with semen quality and freezability.

The PM and TM of the H group were significantly higher than those of the L group, although both groups exhibited similar motility levels before freezing after dilution. This indicates that the freezability of the two groups differed. In the principal component analysis (PCA), the two groups could not be significantly distinguished, as the mean relative standard deviation (RSD) of each group was greater than 0.2. The composition distribution of each sample showed considerable variability, and the correlation heatmap clustering further demonstrated that the differences within groups were greater than those between groups. This suggests that the variability in sperm parameters (SP) was relatively large, and compared to the variation between samples, the differences between groups were minimal. Of the 348 metabolites identified, only 12 differential metabolites were found, representing a relatively small proportion of just 3.4%. This finding is consistent with the results of Xu Bingbing [3,5] in highly and lowly freezable Inner Mongolia cashmere goats, as well as those of Serena Correnti [6] in infertile patients compared to normal individuals. However, other studies have reported a higher proportion of different metabolites in boar sperm with varying freezability. For instance, Heming Sui et al. [7] identified 185 out of 755 differentially expressed metabolites (DEMs), while Sheng Mei [1] identified 185 out of 621. The reasons for these discrepancies may stem from differences in sample selection methods and freezing conditions.

The 348 metabolites identified in SP were predominantly hormones and neurotransmitters, accounting for over 75% of the total, which underscores the significant regulatory role of SP in sperm function. Among these metabolites, glycerophosphate choline, a key component of cell membranes and a precursor of the neurotransmitter acetylcholine was found to be the most abundant. Its presence in SP is closely positively correlated with bull fertility [8]. Although L-carnitine did not show a significant difference between the high (H) and low (L) groups, its correlation with mean progressive motility (PM) at 10 min and 2 h post-thawing approached significance (*p* = 0.051). L-carnitine, an amino acid derivative, facilitates the entry of long-chain fatty acids into mitochondria for β-oxidation. The addition of L-carnitine to the diluent has been shown to enhance preservation effects at room temperature [9] and to improve the cryopreservation of sperm quality in boar semen [10]. Moreover, dietary L-carnitine supplementation has been reported to enhance sperm quality in boars [11]. Acetylcarnitine has also demonstrated the ability to improve semen quality in patients with oligospermia [12,13,14] and stallions [15]. Additionally, incorporating acetylcarnitine during human sperm freezing may reduce oxidative damage [16]. Both glycerophosphocholine and acetylcarnitine could potentially serve as substitutes for egg yolk in human sperm freezing [17]. Other derivatives, such as butyl-L-carnitine, (±)-propionyl carnitine, and (2R)-3-hydroxyl isovaleryl carnitine, are involved in mitochondrial energy metabolism; however, their specific roles in semen remain largely unexplored. Acetylcholine, an important neurotransmitter, is known to elevate sperm calcium levels [18] and regulate sperm motility [19] and other functions. Sphingosine, a crucial component of cell membranes, has yet to be investigated for its specific role in semen, particularly regarding c17-sphingosine.

In both groups of seminal plasma, 12 DEMs were identified, primarily consisting of phospholipids, sphingolipids, peptides, and other compounds. Phosphatidylethanolamine (PE) and sphingomyelin (SM) are essential components of the sperm membrane and play a crucial role in the health and functionality of sperm. Alterations in PE and SM lipid levels in semen may impact the stability of the sperm membrane and its fertilization capability [20,21]. In this study, three PE lipids (PE19:1, PE34:2, PE21:2) and three SM lipids (SM d28:1, SM d32:1, and N-Tetracosanoyl-4-sphingenyl-1-O-phosphorylcholine) were selected for analysis, although their biological functions in semen remain unknown. Additionally, the tripeptide Asp-Asp-Tyr, likely a product of protein degradation, has not been reported for any specific biological function. Other compounds, such as CAY10606, have not been documented in the literature regarding their relationship with semen, indicating that further experiments are necessary to elucidate their biological roles in this context.

Six metabolites were significantly negatively associated with thawing PM, one of which was myristamine oxide, a surfactant that may be a component of cleaning agents used on pig farms. Minoxidil, a treatment for hair loss, has been shown to improve pig hair and could potentially be illicitly added to premixed feed [22]. Additionally, it may be used to treat skin diseases [23]. Octenoyl-carnitine, another carnitine metabolite, has an unknown specific function in semen. Spisulosine is known for its anticancer activity [24], but its role in semen has not been reported. Meanwhile, 2-Hydroxy-4-(octyloxyl)benzophenone is a UV absorber [25] commonly used in sunscreens and other products. The effects of 2-ethylsulfanyl-N-[[1-(hydroxymethyl)cyclopropyl]methyl]benzamide have not been documented. It is important to note that myristamine oxide, minoxidil, spisulosine, and 2-Hydroxy-4-(octyloxyl) benzophenone are all considered environmental pollutants or drug residues. Their presence may be toxic or indicate that the boars have suffered from diseases, potentially leading to a decline in semen quality.

Among the metabolites positively correlated with thawing PM, phospholipids were the most abundant, comprising six of the identified metabolites. Another metabolite was observed at a near-significant level. This abundance may be attributed to the fact that phospholipids are the primary components of cell membranes. Glycerophosphate choline has been previously discussed in this context, while the specific biological functions of the remaining phospholipids in semen remain unknown.

Some oligopeptides may be products of protein degradation. Gly-Gly-Gln (GGQ) is a tripeptide that is essential for peptidyl transferase activity, which is related to peptide chain termination and may influence sperm protein synthesis; however, its specific role in semen has not been reported [26]. Similarly, Asp-Asp-Tyr, mentioned earlier, lacks documented specific biological functions, as do the other two oligopeptides.

Sphingolipids are important components of cell membranes, and the sphingolipids present in semen may be related to the integrity of the sperm plasma membrane. However, the specific biological function of PS 34:2 has not been reported.

Genistein is one of the three aldoketones, an isoflavone mainly derived from soy and certain herbal sources, and it can bind to the estrogen receptor. Its effects on male testicular function and semen quality are controversial; however, it can produce beneficial effects at the appropriate dose, type, route of administration, and timing [27]. In a study where rats were administered 50 mg/day of genistein for 5 days, significant improvements in sperm development were observed, along with increased blood testosterone levels and enhanced semen oxygen consumption [28]. The addition of genistein after thawing can reduce DNA damage in bovine sperm [29]. Furthermore, incorporating genistein during the liquid preservation of human sperm can extend the survival period to 11 days [30]. Additionally, adding genistein to the thawing solution can improve the antioxidant capacity of human frozen semen, increase motility, and reduce plasma membrane damage and DNA damage [31]. 3,4-Dihydroxybenzaldehyde is an antioxidant that mitigates cytotoxicity and oxidative damage induced by fluoride, cadmium (VI) ions, and arsenic (III) in human red blood cells [32,33,34].

Creatine is widely distributed in muscle tissue and, through the action of creatine kinase and magnesium ions, can convert ADP to ATP, providing energy during rapid movements. In sperm, creatine and creatine kinases play a crucial role in regulating sperm quality [35]. In a study involving roosters, feeding the creatine precursor glycocyamine was shown to increase sperm concentration, total sperm count, and motility [36]. Additionally, in the presence of calcium ions and phosphocreatine, human sperm motility and movement speed significantly improved [37]. S-adenosylmethionine, a product of S-adenosine methionine after it loses its methyl group, negatively feedback suppresses methylase activity and is often regarded as a methylation inhibitor [38]. In seminal plasma, S-adenosylmethionine may influence motility and other sperm functions by affecting the sperm methylation process [39,40].

Dodecylmorpholine is commonly used in fungicides and surfactants. Its structure is similar to that of 4-dodecylmorpholine, 4-dodecylmorpholine N-oxide, and N-dodecylmorpholine, which can accumulate in lysosomes and exhibit cytotoxic effects against in vitro cultured tumor cells, potentially affecting ATP synthesis and other cellular functions [41,42]. However, there have been no reported studies on the effects of 4-dodecylmorpholine on sperm.

The RSD, PCA analysis, and correlation heatmap of the samples were consistent with the metabolome, indicating that the differences observed at the proteome level were primarily due to variations among the samples rather than the presence or absence of freezability. The similarities among the samples were found to be much greater than the differences.

Among the proteins with high expression in the H group, the most significantly upregulated was crystallin alpha A (CRYAA), along with its family members CRYBB2 and CRYBB1, which were also upregulated. Crystallins are the main components of the eye lens; however, their impact on sperm development and function remains an area worthy of exploration. The copper transporter CUTC participates in the regulation of copper homeostasis, and its silencing can induce apoptosis in certain cell types [43]. As an essential trace element, copper is involved in numerous physiological and biochemical reactions, particularly in important proteins such as superoxide dismutase. It plays a critical role in the antioxidant functions of sperm, thereby influencing sperm development, motility, and fertilization [44,45]. Profilin 1 (PFN1), also known as actin regulatory protein 1, regulates actin polymerization and depolymerization, which is crucial for maintaining the dynamic balance of the cytoskeleton and motor function. Given that sperm require rapid movement, PFN1 may play significant roles in sperm motility and the fertilization process. Yunxiang Zhao et al. [46] identified PFN1 as a candidate gene affecting sperm morphology in the Duroc boar population through genome-wide association studies (GWAS). Additionally, nicotine has been shown to increase sperm motility in mice [47], which was associated with hypomethylation of the testicular PFN1 gene promoter, suggesting a link between PFN1 and sperm motility.

SHANK1 is a synapse-associated protein generally expressed in the nervous system and is closely related to disorders such as autism [48]. In Drosophila, knockout of SHANK family genes does not affect the reproductive function of either sex [49]; however, its relationship with mammalian sperm function has not been reported. The role of ENPP2, also known as extracellular nucleotide pyrophosphatase or phosphodiesterase 2, in reproductive function remains unclear. ENPP2 can degrade extracellular ATP and ADP to produce AMP, potentially influencing extracellular ATP signaling. Therefore, it can be speculated that ENPP2 may regulate sperm function and affect the fertilization process by modulating the internal environment of the female reproductive tract. Plastin 1 (PLS1) is an actin-binding protein involved in maintaining the stability of the cytoskeleton, which may be critical for sperm flagella formation, maintenance, and motility. In rat testes, PLS3, a member of the same family, has been shown to regulate spermatogenesis [50]. In human follicular fluid, PLS3 is highly correlated with egg quality [51]. However, there is limited research on the correlation between PLS1 and male reproduction. Cylicin 2 (CYLC2) is a sperm-specific protein [52], located on an autosome, while Cylicin 1 is found on the X chromosome. Cylicins are located in the acrosome region of round sperm and the nuclear sheath calyx of mature sperm. Knockout of Cylicin 1 leads to low fertility in male mice, and the knockout of any two copies of Cylicin 1 and Cylicin 2 can result in male infertility. In high-freezability bull sperm [53], CYLC2 protein was highly expressed, consistent with the findings of the present study. In contrast, Huntington interacting protein 1 (HIP1) was highly expressed in low-freezability sperm, which is contrary to the results of this study. However, in asthenozoospermic sperm, HIP1 mRNA expression was found to be decreased compared to controls [54], indicating a positive correlation between HIP1 and human sperm quality. The relationship between phospholipase A1 member A (PLA1A) and sperm has not been reported; however, its involvement in phospholipid metabolism may play an important role in maintaining sperm plasma membrane integrity and facilitating the acrosome reaction. Cathepsin B (CSTB) was found to be more highly expressed in the medium of successfully fertilized embryos compared to those that were not fertilized, suggesting that CSTB may be related to embryo implantation. At 5 °C, CSTB and legumain (LGMN) in semen with high and low cold tolerance exhibited significant differences, indicating that CSTB and LGMN may be associated with the low-temperature tolerance of boar sperm.

The protein with the lowest expression, cell communication network factor 6 (CCN6), has not been reported in association with male reproductive function. Apolipoprotein A2 (APOA2), a primary component of high-density lipoprotein (HDL), is involved in the transport and metabolism of fats and cholesterol. In Bali bulls, APOA2 may be related to fertility [55]. In males with varicocele-related infertility [56], APOA2 has been associated with testicular oxidative stress and sperm function, and its expression was significantly downregulated [57] in the seminal plasma of patients with varicocele. Furthermore, in mice, the circadian rhythm of apolipoproteins, including APOA2, can influence the production of testicular testosterone. Tumor-associated calcium signal transduction protein 2 (TACSTD2), which is generally highly expressed in tumors and cancer cells, is also found in embryos; however, reports on its association with spermatogenesis and sperm function are scarce. Serum amyloid A3 (SAA3), an inflammatory marker [58,59], may indicate that boars are undergoing inflammation, which can lead to reduced sperm quality and affect freezing resistance. Sialidase 1 (NEU1) can impact fertilization [60] by facilitating sialylation [61] during capacitation. CCR4-non-transcription complex subunit 7 (CNOT7) is an important protein in spermatogenesis; its knockout in mice results in morphologically abnormal sperm [62]. Additionally, Dai et al. [63] found that CNOT4 and other CCR4 subunits, including CNOT7, are involved in mRNA degradation, efficient DNA damage repair, and XY chromosome crossover during meiosis in male germ cells. Adipogenesis regulatory factor (ADIRF) has limited reports on reproductive function but is associated with various diseases. For instance, ADIRF protein in urine exosomes can serve as a biomarker for prostate cancer [64]. Hypermethylation of the ADIRF promoter regulates its expression level and participates in NNK-induced malignant transformation of pulmonary bronchial epithelial cells [65]. The function of Suprabasin (SBSN) has long been unclear, although some studies [66] have suggested its association with cancer and other diseases. The relationships between ectodysplasin A (EDA), zymogen granule protein 16B (ZG16B), and spermatogenesis and function remain unclear. However, both proteins are linked to specific diseases [67,68,69,70], and their elevated levels may indicate potential health issues in boars, which could indirectly affect sperm quality and freezability.

The most prominent item for gene ontology biological process (GO_BP) enrichment was GO:0032232, which pertains to the negative regulation of actin filament bundle assembly. The three proteins involved in this process are SHANK1, TACSTD2, and Profilin 1 (PFN1). As discussed previously, the roles of SHANK1 and TACSTD2 in reproductive processes remain unclear. However, both SHANK1 and PFN1 are known to negatively regulate actin filament bundle assembly. Additionally, TACSTD2 and PFN1 are implicated in the negative regulation of GO:0051497, which relates to stress fiber assembly. It is speculated that SHANK1 and TACSTD2, similar to PFN1, may be involved in the morphogenesis of sperm flagella during spermatogenesis and could play important roles in maintaining the motility of mature sperm.

Furthermore, in addition to PLS1, which is bound to actin, the number of proteins related to motility structures reached at least four. This underscores the significant influence of seminal plasma (SP) on sperm motility.

In terms of MF, four out of the eight most significant items were related to cell adhesion, involving the proteins EPCAM (epithelial cell adhesion molecule), CCN6, TACSTD2, and VTN (vitronectin). Among these, EPCAM is known to play a role in spermatogonial stem cell differentiation and spermatogenesis [71]. Normal mature sperm are typically motile and free; however, sperm agglutination is associated with cell adhesion in semen. These adhesion-related proteins may contribute to sperm agglutination and are therefore negatively correlated with semen quality. Nevertheless, during spermatogenesis and fertilization, sperm interact with Sertoli cells, the female reproductive tract, and the egg, in which cell adhesion molecules play a crucial role.

Considering only a significance level of *p* < 0.05, the number of DEPs identified was 65, which is close to the number of PAMPs at 70. However, due to the differing analytical approaches, the two results remain notably distinct. Among the significantly related proteins with small fold changes (FC), there were nine upregulated proteins: α-L-fucosidase 2 (FUCA2), legumain (LGMN), pyruvate kinase M (PKM), alpha-N-acetylgalactosaminidase (NAGA), interleukin 1 receptor 1 (IL1R1), neuroplastin (NPTN), and serine protease 8 (PRSS8). In addition, there were two downregulated proteins: cathepsin S (CTSS) and A0A480K168.

FUCA, an enzyme capable of hydrolyzing fucosidic bonds on glycoproteins, exhibits different isoforms across various species in semen. In bulls, two types of FUCA are present in the testis and epididymis: FUCA1 and FUCA2. While both isoforms are found in the testis and epididymis, FUCA1 is the only isoform detected in the epididymis, sperm, and seminal plasma (SP) [72]. Notably, the concentration of FUCA in the epididymal fluid of high-fertility bulls is significantly higher than that in low-fertility bulls [73]. Furthermore, purified FUCA from bull SP has been shown to promote the acrosome reaction in mouse sperm [74]. In human semen, various FUCA isoforms [75,76] are present in SP and on sperm, primarily localized to the posterior region of the sperm head. In mice [77], one type of FUCA is found in the epididymal tail, while two isoforms are present in sperm. The epididymal tail fluid contributes the majority of FUCA activity; during the acrosome reaction, soluble fucosidase is released outside the cell, while a small amount of membrane-bound FUCA persists until the completion of the acrosome reaction. FUCA also plays a critical role in sperm–egg fusion in hamsters [78] and is involved in promoting sperm capacitation in pigs [79]. However, there is a paucity of studies exploring the relationship between FUCA and sperm freezing resistance.

LGMN is present in the SP of bulls and other livestock species. Notably, LGMN levels in boar SP have been positively correlated with the tolerance of sperm to autologous SP [4]. Additionally, both LGMN and TSB exhibit high expression levels in boars that are resistant to storage at 5 °C [80]. This relationship between SP tolerance and low-temperature tolerance may contribute to the observed positive correlation with sperm freezability. In bulls, SP LGMN has also been positively correlated with the rate of acrosome-intact live sperm in fresh semen [81]. Furthermore, LGMN is implicated in the synthesis of cathepsins. Kanae et al. [82] reported that LGMN-deficient mice exhibited weight loss but maintained normal fertility. However, these mice displayed abnormal lysosomal function, with inactive lysosomal enzymes and impaired synthesis of cathepsins B, H, and L.

Pyruvate kinase (PK) is a key enzyme in glycolysis that catalyzes the conversion of phosphoenolpyruvate to pyruvate, facilitating the transfer of high-energy phosphate bonds to ADP, resulting in the production of ATP. PKM is a specific isoform of pyruvate kinase. Conditional knockout studies in mice [83] have shown that while germ cells can produce morphologically normal sperm, the resulting sperm exhibit reduced motility and ATP levels, leading to infertility. This suggests that PKM is not essential for spermatogenesis in mice but is crucial for maintaining sperm motility and fertilization capacity. Exosomes [84] have been shown to synthesize ATP in vitro via the glycolytic pathway, indicating that PKM may play a role in regulating sperm function either through the fusion of exosomes with sperm or by acting as a signaling molecule. Another isoform, PKS, is found in the acrosome and fibrous sheath region [85] of boar sperm, suggesting that PK may also be involved in the acrosome reaction.

There are few studies on the role of NAGA in sperm and semen. However, Nagdas et al. [86] found that NAGA can bind to the outer acrosome membrane-associated matrix complex (OMC), suggesting that NAGA may be involved in the sperm acrosome reaction, sperm-egg recognition, and other related processes.

NPTN is primarily expressed in the nervous system but is also essential for reproductive behavior. Mice with a total knockout of NPTN produce normal amounts of sperm; however, their blood testosterone levels remain consistently low, akin to those of juvenile mice, rendering the knockout mice sterile [87].

PRSS8 is a serine protease that plays a crucial role in maintaining skin integrity and facilitating the spread of cancer cells. In the context of reproduction, Yu et al. [88] reported that PRSS8 levels were significantly reduced in the exosomes of mice with asthenozoospermia, indicating that PRSS8 may be involved in the spermatogenesis process.

Inayat et al. [89] found that the concentration of CTSB in human seminal plasma was approximately 70 times higher than that in serum samples, while the levels of CTSL in seminal plasma were about 500 times greater, and CTSS levels were around 40 times higher. These three cathepsins were also found to be bound to the surface of prostasomes, suggesting that the high content of cathepsins in seminal plasma plays a significant role. E. B. Menezes et al. [81] reported that the quality of fresh bull semen was highly positively correlated with clusterin (CLU), tissue inhibitor of metalloproteinases 2 (TIMP2), CTSS, and legumain (LGMN) in seminal plasma. Additionally, the thawing motility of frozen semen was positively correlated with CLU levels. Abdullah Baharun et al. [90] observed high expression of CTSS in semen of extremely poor quality or bulls lacking sperm, indicating that CTSS is negatively correlated with semen quality.

SP functions as an extracellular fluid but is enriched with various intracellular anatomical structures, including cytoplasm, organelles, and cell fluids, which are reflected in its subcellular localization. In addition to cells, SP contains components derived from cytoplasmic cell parts, such as the nucleus, which may be present in cellular debris, vesicles, and exosome secretions. Many substances can enter sperm cells through the fusion of exosomes and vesicles with the sperm membrane. These intracellular substances may play unique roles in the composition and function of seminal plasma.

Six of the GO_BP items identified were related to cell cycle and division. In the context of sperm, only spermatogenesis is directly involved in these processes, suggesting that freezing resistance may be significantly influenced by the spermatogenesis process. Furthermore, the KEGG pathway analysis highlighted pyruvate metabolism and heat production, both of which are associated with energy production. The combined analysis of the proteome and metabolome also indicated that arginine and proline metabolism, along with heat production, are related to energy production. These findings underscore the critical role of energy production in determining freezing resistance.

In our study, we applied SP as a basic freezing liquid and found that the PM, TM, and movement parameters in the high (H) group were higher than those in the low (L) group; however, these differences did not reach statistical significance. This indicates that the protective effect of SP on sperm freezing is quite limited. Furthermore, when compared to the control group using a commercial freezing extender, no significant differences were observed between the H and L groups, and both groups exhibited poorer performance two hours post-thawing. This suggests that SP may not be the optimal choice for a freezing solution. Additionally, the PM and TM values at 10 min in Figure 7 were significantly higher than those presented in Figure 1. This variation may be attributed to seasonal effects and batch differences, as the experiments depicted in Figure 7 were conducted in cooler October and November, while those in Figure 1 were conducted in hotter June.

Given the multitude of factors influencing sperm freezability, it is important to recognize that seminal plasma (SP) constitutes only one component among many. Therefore, it should not be anticipated that SP will exert a decisive protective effect on sperm. While numerous metabolites and proteins associated with freezability in semen are linked to male reproduction, their specific roles in sperm remain unclear. Further research is needed to elucidate how these substances impact freezability, as they may play a significant role in enhancing sperm viability during freezing.

## 5. Conclusions

The variability of SP metabolites and proteins is notably greater between samples than among different freezable groups, indicating that the overall similarity among samples exceeds the differences observed.

Notably, SP with high and low freezability differs significantly in 12 metabolites, including several phosphatidic acid ethanolamines, and in 38 proteins such as CRYAA, CUTC, SHANK1, PFN1, NEU1, SAA3, TACSTD2, APOA2, and CCN6, all of which are involved in the regulation of biological processes and hydrolytic enzyme activity. Additionally, we identified 33 metabolites associated with antimicrobial peptides (MAMPs), including glycerol phosphocholine and creatine, as well as 70 metabolites associated with pathogen-associated molecular patterns (PAMPs), such as LGMN and CTSB. These PAMPs are implicated in cell cycle regulation, division, and pyruvate metabolism. The combined analysis of the metabolome and proteome revealed that both MAMPs and PAMPs are involved in thermogenesis, arginine–proline metabolism, and ether–ester metabolism. However, the specific roles of many proteins and metabolites in spermatogenesis, sperm function, and fertilization remain unclear and warrant further experimental investigation.

Furthermore, the protective effect of SP with high freezability on sperm is not significantly stronger than that of SP with low freezability. When compared to commercial semen freezing reagents, SP does not demonstrate superior cryoprotective properties.

## Figures and Tables

**Figure 1 animals-14-03656-f001:**
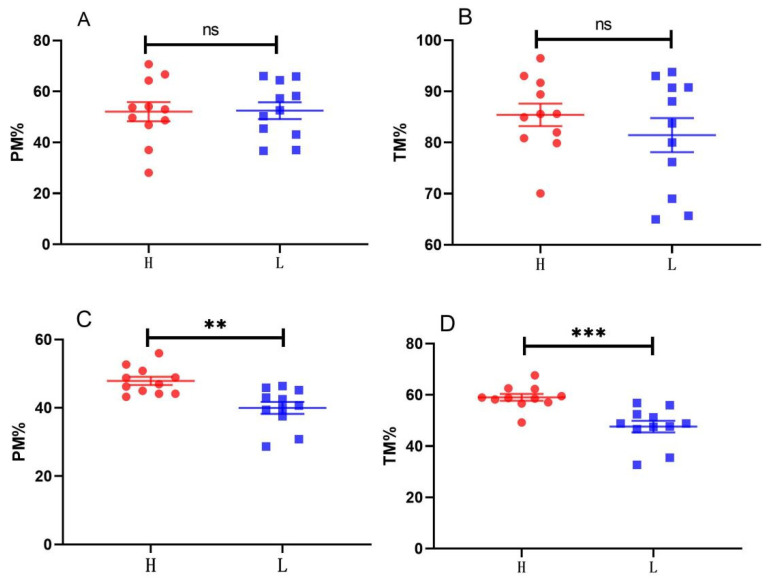
Comparison of the motility of highly and lowly freezable groups before and after freezing. (**A**,**B**) PM and TM of groups H and L after pre-diluted before freezing. (**C**,**D**) PM and TM of groups H and L after thawing. H, high freezability. L, low freezability. PM, progress motility. TM, total motility. “ns”, not significant. “**” “***” means significant difference and “*p* < 0.01” “*p* < 0.001”, respectively. Sample size *n* = 11.

**Figure 2 animals-14-03656-f002:**
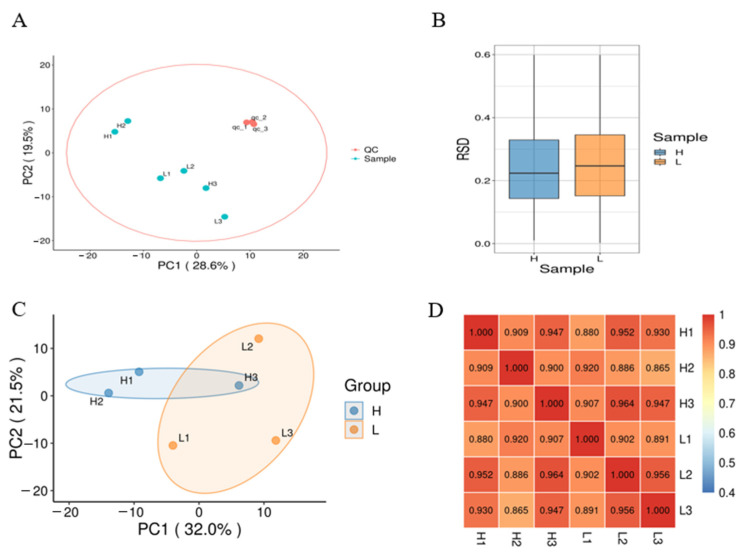
Quality control and repeatability analysis of metabolome of highly and lowly freezable seminal plasma. (**A**) Principal component analysis (PCA) including samples and quality control group. (**B**) The relative standard deviation of samples. (**C**) PCA of group H and L. (**D**) Heatmap of relativity between samples. H, high freezability; L, low freezability. H1,H2,H3, samples of group H;L1, L2, L3, samples of group L. QC, quality control. Each sample was mixed with 3 boar seminal plasma samples with equal volume.

**Figure 3 animals-14-03656-f003:**
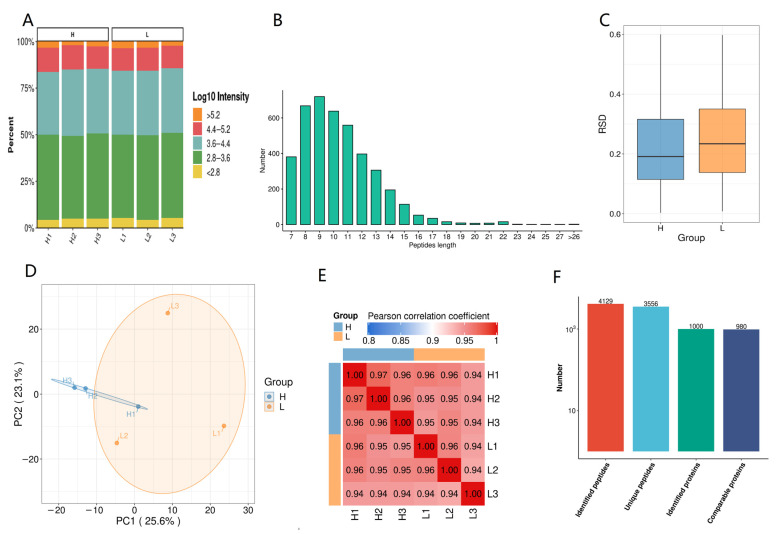
Quality control and reproducibility analysis of the proteome of highly and lowly freezable seminal plasma. (**A**) Signal intensity distribution of samples. (**B**) Distribution of peptide lengths. (**C**) RSD of group H and L. (**D**) PCA of group H and L. (**E**) Heatmap of relativity between samples. (**F**) Number of identified peptides and proteins. H, high freezability group; L, low freezability group. H1, H2, H3, samples of group H; L1, L2, L3, samples of group L. Each sample was mixed with 3 boar seminal plasma samples with equal volume.

**Figure 4 animals-14-03656-f004:**
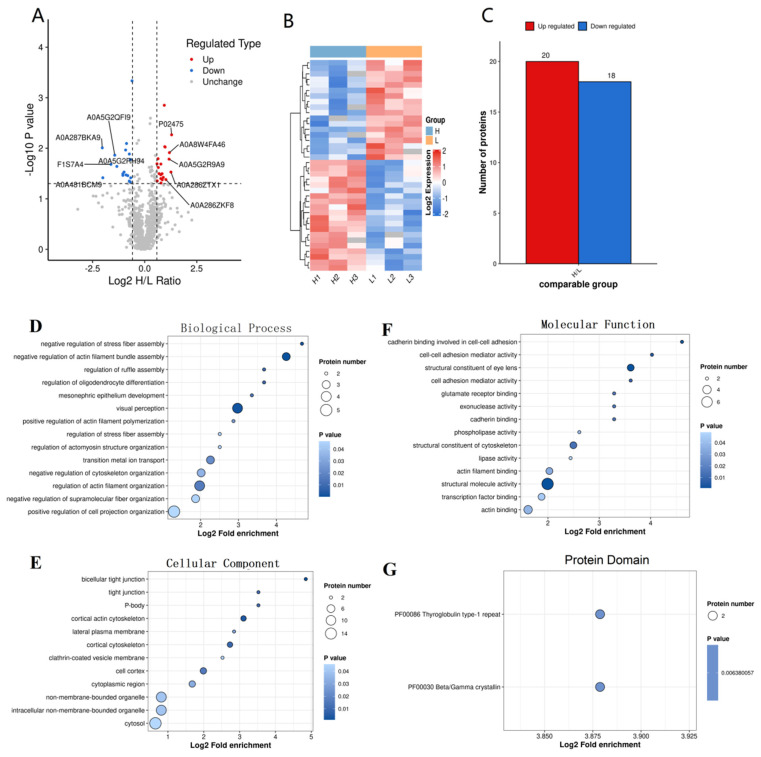
Differential expressed proteins (DEPs) analysis and GO enrichment. (**A**) Volcano plot for DEPs screening. The top 5 upregulated proteins (URPs) (red dots) and downregulated proteins (DRPs) (blue dots) are labeled with Uniprot IDs. The thresholds of fold_change were set as 1.5 and 1/1.5. *p*-value < 0.05 was defined as statistically significant. (**B**) Heatmap of DEP expression levels of each sample. (**C**) Number of regulated proteins. (**D**) GO_BP enrichment of DEPs. (**E**) GO_CC enrichment of DEPs. (**F**) GO_MF enrichment of DEPs. (**G**) Protein domain enrichment of DEPs. H, high freezability group; L, low freezability group. H1, H2, H3, samples of group H; L1, L2, L3, samples of group L. Each sample was mixed with 3 boar seminal plasma with equal volume.

**Figure 5 animals-14-03656-f005:**
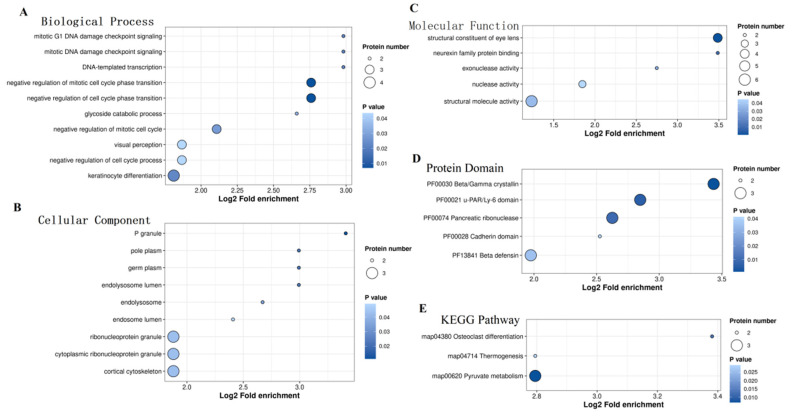
GO and KEGG pathway enrichment of proteins associated with mean PM (PAPMs) of 10 min and 2 h after thawing. (**A**) GO_BP enrichment of PAPMs. (**B**) GO_CC enrichment of PAPMs. (**C**) GO_MF enrichment of PAPMs. (**D**) Protein domain enrichment of PAPMs. (**E**) KEGG pathway enrichment of PAPMs.

**Figure 6 animals-14-03656-f006:**
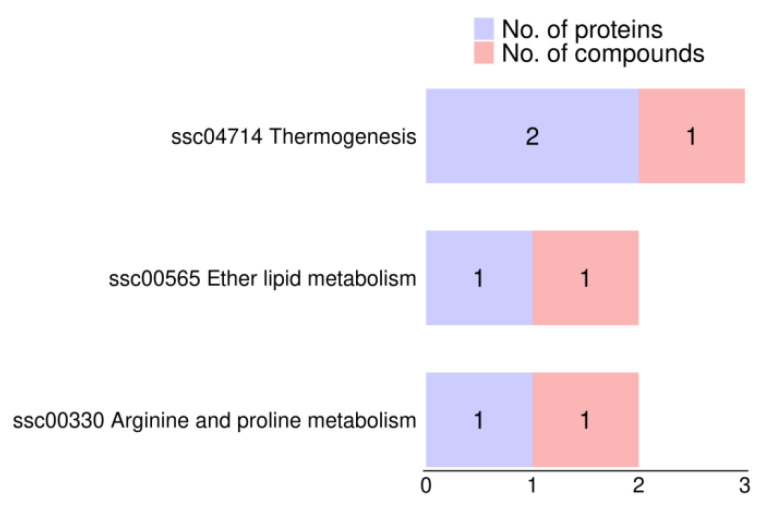
KEGG pathway enrichment in a combination of proteins and metabolites associated with mean PM of 10 min and 2 h after thawing.

**Figure 7 animals-14-03656-f007:**
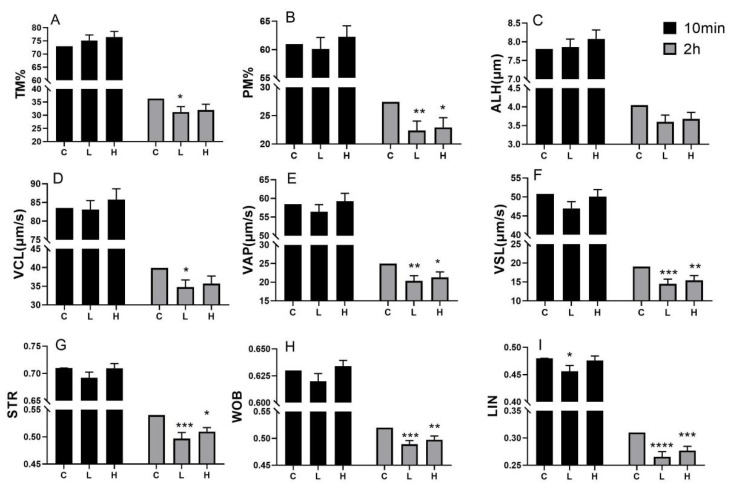
Effect of seminal plasma as base freezing extender on thawing motility and movement parameters of frozen semen. (**A**–**I**): TM, PM, ALH, VCL, VAP, VSL, STR, WOB and LIN at 10 min and 2 h after thawing of groups with different base freezing extender. H, high freezability seminal plasma (SP) group. L, low freezability SP group. PM, progress motility. TM, total motility. VCL, velocity of curve line. VSL velocity of straight line. VAP, velocity of path rate. ALH. Amplitude of lateral head. STR, straightness. WOB, wobble. LIN, linear. Individual boar differences were removed for all data. *n* = 20. “ns”, not significant. “*” “**” “***” “****” means significant difference with control and “*p* < 0.05” “*p* < 0.01” “*p* < 0.001” “*p* < 0.0001”, respectively.

**Table 1 animals-14-03656-t001:** Correlation between reproductive hormone in SP and motility before and after freezing.

		Pre-Diluted PM	Pre-Diluted TM	Original Concentration	10 min PM	10 min TM	2 h PM	2 h TM
T	*r*	0.087	−0.066	−0.241	0.074	0.133	−0.163	−0.114
	*p*	0.636	0.718	0.183	0.692	0.476	0.456	0.606
P	*r*	−0.341	0.097	0.177	−0.044	−0.058	0.063	0.03
	*p*	0.056	0.598	0.334	0.815	0.759	0.776	0.892
LH	*r*	−0.105	−0.17	−0.14	−0.073	−0.069	−0.153	−0.212
	*p*	0.568	0.352	0.445	0.698	0.712	0.486	0.331
PRL	*r*	0.237	0.231	0.128	−0.104	−0.072	−0.182	−0.177
	*p*	0.191	0.203	0.486	0.577	0.699	0.406	0.42
E2	*r*	0.146	0.068	0.238	−0.229	−0.205	−0.361	−0.35
	*p*	0.427	0.712	0.19	0.216	0.27	0.091	0.101
PGF2α	*r*	0.17	0.062	0.217	−0.109	−0.118	−0.272	−0.301
	*p*	0.353	0.734	0.232	0.56	0.527	0.209	0.162
T/E2	*r*	0.034	−0.086	−0.321	0.142	0.196	−0.053	−0.002
	*p*	0.855	0.638	0.073	0.446	0.291	0.811	0.991
P/E2	*r*	−0.303	0.076	0.079	0.065	0.046	0.245	0.213
	*p*	0.091	0.681	0.665	0.727	0.807	0.259	0.33
	*n*	32	32	32	31	31	23	23

Note: SP, seminal plasma; PM, progress motility; TM, total motility. *r*, the Pearson correlation coefficient; *p*, the *p*-value (two-tailed) of *r*; *n*, the sample size.

**Table 2 animals-14-03656-t002:** List of DEMs between highly and lowly freezable SP.

Index	Compounds	H/L FC	H/L *p*-Value	H/L VIP
PTM_606	PE 19:1	1.913	0.0010	2.107
PTM_116	2-ethylsulfanyl-N-[[1-(hydroxymethyl)cyclopropyl]methyl]benzamide	0.570	0.0140	1.985
PTM_282	Asp Asp Tyr	1.804	0.0162	1.942
PTM_739	PS 34:2	7.123	0.0184	1.940
PTM_761	SM d28:1	1.936	0.0201	1.890
PTM_763	SM d32:1	2.160	0.0245	1.896
PTM_552	N-cyclopentyl-3-[4-(5-cyclopropyl-1,2,4-oxadiazol-3-yl)phenoxy]pyrrolidine-1-carboxamide	2.738	0.0261	1.854
PTM_610	PE 21:4	2.733	0.0282	1.906
PTM_567	N-Tetracosanoyl-4-sphingenyl-1-O-phosphorylcholine	8.767	0.0324	1.850
PTM_72	1-Stearoyl-2-linoleoyl-sn-glycero-3-phosphoethanolamine	0.287	0.0345	1.819
PTM_45	1-Palmitoyl-2-lauroyl-sn-glycero-3-phosphorylcholine	1.673	0.0353	1.847
PTM_614	PE 23:5	0.264	0.0447	1.749

DEMs, differentially expressed metabolites. SP, seminal plasma. FC, Fold_Change. PE, phosphoryl ethanolamine; PS, phosphatidyl serine; SM, sphingomyelin.

## Data Availability

The authors confirm that the original contributions presented in this study are reflected in this manuscript.

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
