# Peer review of "Effect of Seminal Plasma on the Freezability of Boar Sperm"

_animals, 2024, doi:10.3390/ani14243656_

Round 1
Reviewer 1 Report
Comments and Suggestions for Authors
This work analyzes the components of SP in good and bad freezer boars to identify the differences that may influence the freezability of spermatocytes. The experimental work related to the detection and identification techniques of metabolites and proteins is very complete and complex. 348 metabolites and 4129 peptides are described. 6 reproductive hormones are also included in the study. The discussion is extensive, but well-crafted. A broad review and contrast is made of the function and possible effects of many of the metabolites and proteins found in SP.
My first doubt is in the experimental design. In the M&M section, it is not described how or when the selection of males is carried out, nor how many animals there are in each experimental group. Are the sperm analyses individual or are 3 males grouped together as in the metabolite analyses?. And my main question, if in the freezing protocol a wash is performed and the supernatant is removed, the SP is also removed with the supernatant. So how does the SP influence the freezing of the semen? Only during the equilibration time at 17ºC.
It does not describe in the M&M section the experiment of using SP as base freezing extender. How do you get SP?, Are the control group ejaculates from high or low freezing?, Have you not thought about using high SP with L sperm?. L318-320 move to M&M section.
As described in M&M, semen samples are diluted and taken to the laboratory immediately after they are obtained. Then a 1 mL of semen was extracted for detecting the original concentration, Progress motility (PM) and total motility (TM). Why do the authors say that these are pre-diluted values? It would be better to say fresh or initial values?
I am unable to find the hormone analysis kits used on the internet, not even the company. Could you provide commercial references? Is there a company website… Beijing Tianyuanaorui Biotechnology Co., Beijing Jinhaikeyu Biology.
Other comments
- L64: Replace ELASA by ELISA
- L94: It is missing to include that the analysis is also done after 2 hours.
- L95-96: It is not necessary to comment on the number of people involved in the analysis, it is assumed that it will be done within a reasonable and effective time.
- L103: what are the movement parameters tested?
- L111: indicate what OD means
- L178: The authors refer for the first time to two groups, but do not define which two groups. At this point we do not know which ones they are.
- L181: change EDMs by DEMs
- L209: Semen quality index? In M&M does not specify any semen quality index.
- L208: missing closing parenthesis
- L209: Why are the parameters T/E2 and P/E2 studied? Why are they not referred to in the M&M section?
- L201: words are repeated. Rewrite sentence
- L225: Here I assume that reference is made to table 2, not table 1. There are 13 DEMs in this table, not 12.
- L279: space between “binding.Only”.
- L541: change the "," to the "."
- Fig 7. indicates the differences with "*", but it is not known if they are with the control or between experimental groups. It should be indicated in some way between which groups there are differences.
Author Response
Q1: My first doubt is in the experimental design. In the M&M section, it is not described how or when the selection of males is carried out, nor how many animals there are in each experimental group. Are the sperm analyses individual or are 3 males grouped together as in the metabolite analyses?. And my main question, if in the freezing protocol a wash is performed and the supernatant is removed, the SP is also removed with the supernatant. So how does the SP influence the freezing of the semen? Only during the equilibration time at 17ºC.
Explain: In the begining of the experiments, the semen of about 30 boars were frozen, 2~4 times for each boar in about half a month. Then the boars were sorted by the mean progress motility (PM) of 10 min and 2h after thawed in each cycle. If the boar ranked in top/bottom 50% at least twice, then the boar was selected as high/low-freezability group. If twice top and twice bottom, neither. Then 11 boars were selected as high freezability and other 11 as low-freezability.
Sperm analyses were done individual. The PM or TM data used in correlation analysis with metabolites or proteins were the mean of the same semen samples.
The sample size of fig1 is 11, fig7 is 20.
Just during the equilibration time at 17ºC can make a difference, that’s why both Minitube and tianyuanaorui have special dilutions for equilibration at 17℃ in their freezing kits. Metabolic rate is positively correlated with the temperature, the time SP treat sperms is short but the temperature is high, so the influence of SP may not be ignored. For the mechanism, it may relate to oxidation resistance and other ways.
Q2 It does not describe in the M&M section the experiment of using SP as base freezing extender. How do you get SP?, Are the control group ejaculates from high or low freezing?, Have you not thought about using high SP with L sperm?. L318-320 move to M&M section.
At the same time when the boars were being selected, all the SP was fozen at -80℃ for use. So, when the boars have been selected, the SP also been selected. All the group ejaculates from both high and low freezing, just the treat was different. We did thought about using high SP with L sperm, but data showed no significant, so we did not dived the data into several figures.
Q3 As described in M&M, semen samples are diluted and taken to the laboratory immediately after they are obtained. Then a 1 mL of semen was extracted for detecting the original concentration, Progress motility (PM) and total motility (TM). Why do the authors say that these are pre-diluted values? It would be better to say fresh or initial values?
Because the dilution was not for liquid storage, the values was not the original semen or as semen diluted by BTS which almost not inhibit sperm movement. The values was lower than fresh semen in fact. The process for semen freezing, there are one-step method and two-step method, all the steps are for freezing dilution. But for boars, there is another step before freezing dilution, so the step is called pre-dilute. So we say these are pre-diluted values.
Q4 I am unable to find the hormone analysis kits used on the internet, not even the company. Could you provide commercial references? Is there a company website… Beijing Tianyuanaorui Biotechnology Co., Beijing Jinhaikeyu Biology.
Beijing Tianyuanaorui Biotechnology Co.: http://www.tianyuanaorui.com
I don’t find a company website of Beijing Jinhaikeyu Biology , but there is a email address jinhaikeyu@sohu.com
Q5:Why are the parameters T/E2 and P/E2 studied? Why are they not referred to in the M&M section?
Because the function of T and E2, P and E2 are contrast in many aspects. So their ratio may be important. The two the parameters T/E2 and P/E2 are so simple derivative variables, and there were no significance, so we think it is not necessary to mention it separately.
Reviewer 2 Report
Comments and Suggestions for Authors
Dear authors,
After carefully reading the manuscript entitled “Effect of seminal plasma on the freezability of boar semen” I’d like to report my review as follow;
In brief, semen samples from boars were cryopreserved and classified into either good or poor freezabilty according to their post-thawed motility results. The seminal plasma from both good and poor freezability semen was examined with proteomics and metabolomics analyses as well as reproductive hormones. Differences in proteomics and metabolomics between two groups of seminal plasma were found whereas no difference in reproductive hormone was detected.
Strength: the manuscript provided valuable information for protein and metabolite characteristics of seminal plasma obtained from good and poor freezability semen. Compared to previous studies, a large number of proteins were identified in this manuscript. Some proteins and metabolites were identified as potential markers of good cryotolerance spermatozoa for further studies.
Weakness: although differences in compositions of seminal fluid were detected, the seminal fluid by itself plays relatively small role in cryotolerance of spermatozoa. Spermatozoa are exposed to seminal fluid for a short period of time since spermatozoa were removed from seminal fluid via centrifugation prior to freezing process. Therefore, molecules or substances identified in the present study might not be useful to predict the freezability of boar semen.
Specific comments
Title
The title of the manuscript is too general. Importantly, there was no clear effect of seminal fluid on freezability of boar semen. The term “characterization of seminal plasma” or similar might be more suitable for the main results of this manuscript.
Abstract
Line 25-26; please avoid using ambiguous wording “slightly higher than….without significant”.
Introduction
To date, proteomics and metabolomics studies are not new, it would be better to include some previous studies into the introduction as well as, if possible, provide some information on the gap of knowledge or why this study is needed.
Materials and Methods
Please explain in detail on how to classify the semen into high or low freezability. The authors stated in the discussion (line 344-347) that it was not ranked according to arithmetic mean of freezing results and also stated that the screening method was effective.
Line 85: please be specific about “good quality” such as % sperm motility and sperm concentration.
Line 86: “balanced”
Line 88: 17ºC to 4ºC for 90 min, was this equal to 0.14ºC/min
Line 115: please provide detail information on how seminal plasma samples were collected and prepared such as collected immediately after semen collection or else, immediately separated from spermatozoa or else.
Results
Line 197-198; the results of 2h after thawed are missing
Figure 1: please use the same scale of Y-axis in all panels.
Line 225: there were 13 not 12 selected DEMs shown in Table 2
Line 245: list of DEMs are supposed to be presented according to their fold changes from top to bottom or their P-values.
Line 318-340 and Figure 7: the results of this experiment made this manuscript less interesting. This experiment draws attentions away from proteomics and metabolomics findings. It was not surprise that seminal plasma can be used to prepare freezing extender because protection against cold shock was provided by egg yolk and protection against freezing damage was provided by glycerol. The reason that SP based freezing extender performed poorly at 2 h may be because of lacking appropriate buffering component in SP based extender compared to a commercial one. Interestingly, the motility results shown in figure 7 were exceptionally higher than those reported in figure 1, please kindly provide some discussion.
Discussion
Compared to the previous study (Sui et al., 2023), only 348 metabolites were identified in the present manuscript. Please kindly provide some discussion.
Overall recommendation
I’d like to recommend this manuscript as Minor Revisions.
Author Response
Comments 1: The title of the manuscript is too general. Importantly, there was no clear effect of seminal fluid on freezability of boar semen. The term “characterization of seminal plasma” or similar might be more suitable for the main results of this manuscript. |
Response 1:Thank you for pointing this out. I/We agree with this comment. However, The whole experiment was based on high and low freezing resistance, and all the different metabolites and proteins were related to freezing resistance. Although the final effect of freezing the base liquid was not obvious, in general, we thought that the original topic was more appropriate |
Comments 2: Line 25-26; please avoid using ambiguous wording “slightly higher than….without significant”. |
Response 2: Agree. I/We have, accordingly, done/revised/changed/modified…..to emphasize this point. When using highly and lowly freezable SP as base freezing extenders, the protective effect of highly freezable SP was not significantly superior to that of lowly freezable SP, and it did not outperform the control group using a commercial freezing extender. |
Comments 3: To date, proteomics and metabolomics studies are not new, it would be better to include some previous studies into the introduction as well as, if possible, provide some information on the gap of knowledge or why this study is needed. |
Response 3: The proteomics and metabolomics studies on SP are not too much, Especially for freezing resistance, the mainly highly relevant references have been cited |
Comments 4: Please explain in detail on how to classify the semen into high or low freezability. The authors stated in the discussion (line 344-347) that it was not ranked according to arithmetic mean of freezing results and also stated that the screening method was effective. Line 115: please provide detail information on how seminal plasma samples were collected and prepared such as collected immediately after semen collection or else, immediately separated from spermatozoa or else. |
Response 4:Agree.Has added a section specifically to explain how to operate. 2.2.4. grouping of boars and separation of SP At the beginning of the experiments, semen from approximately 30 boars was frozen 2 to 4 times for each boar over a period of about half a month. The boars were then sorted based on the mean progress motility (PM) measured at 10 minutes and 2 hours after thawing during each semen collection cycle. A boar was classified into the high or low freezability group if it ranked in the top or bottom 50% at least twice. If a boar ranked in the top twice and in the bottom twice, it was not classified into either group. Ultimately, 11 boars were selected as high freezability and another 11 as low freezability. Concurrently, all the seminal plasma (SP) was separated by centrifugation at 1000 rpm for 13min, then filtered through 3μm, 1μm, and 0.22μm PVDF filters, and frozen at -80°C for later use. Thus, when the boars were selected, the corresponding SP was also selected. |
Comments 5:Line 85: please be specific about “good quality” such as % sperm motility and sperm concentration. |
Response 5:Semen samples with a concentration greater than 2 × 10^8 and an abnormal rate of less than 10% were selected for the experiment and equilibrated at 17°C for 1 hour. |
Comments 6:Line 197-198; the results of 2h after thawed are missing |
Response 6:The results was the average results of 10 min and 2h after semen thawing. However, for the average results 10 min and 2h after semen thawing, the PM of the H group was significantly higher (p < 0.01) than that of the L group (49.81% ± 1.04% vs. 39.57% ± 1.62%), while the TM showed an extremely significant difference (p < 0.001) (59.03% ± 1.3% vs. 47.53% ± 2.16%). |
Comments 7:Line 225: there were 13 not 12 selected DEMs shown in Table 2 |
Response 7:The number has been corrected. |
Comments 8: Line 245: list of DEMs are supposed to be presented according to their fold changes from top to bottom or their P-values. |
Response 8:the list of DEMs are presented according to their P-values |
Comments 9: Line 318-340 and Figure 7: the results of this experiment made this manuscript less interesting. This experiment draws attentions away from proteomics and metabolomics findings. It was not surprise that seminal plasma can be used to prepare freezing extender because protection against cold shock was provided by egg yolk and protection against freezing damage was provided by glycerol. The reason that SP based freezing extender performed poorly at 2 h may be because of lacking appropriate buffering component in SP based extender compared to a commercial one. Interestingly, the motility results shown in figure 7 were exceptionally higher than those reported in figure 1, please kindly provide some discussion. |
Response 9:The major component of control is lactose,also lacking appropriate buffering component. All the major buffering component are proteins in egg yolk. Although the main results are metabolome and proteome, there are both components in the metabolome and proteome that improve freezing resistance and are not conducive to freezing resistance. The overall freezing effect still needs to be verified by experiments. Therefore, we designed this experiment. Follow-up studies can try to increase the content of antifreeze components or reduce the adverse antifreeze components. Additionally, the PM and TM values at 10 min in Figure 7 were significantly higher than those presented in Figure 1. This variation may be attributed to seasonal effects and batch differences, as the experiments depicted in Figure 7 were conducted in cooler October and November, while those in Figure 1 were conducted in hotter June. |
Reviewer 3 Report
Comments and Suggestions for Authors
The article presented here is describing an experiment related to a proteomic and metabolomic approach on a group of low (L) and high (H) motility frozen-thawed boar frozen semen. unfortunately, it is quite hard to follow and understand the description presented by the authors due to a low level of explanation and maybe to a too low level of English in this paper. Here are some principal points that, once adjusted together with the English check, may help in the comprehension of the paper.
1. Just at the beginning of the discussion, we discover the number of Boars included in the study. that should be clearly stated and moved in M&M, together with an explanation of how they have been divided in groups and how many ejaculates per each animal have been collected.
2. It is not clear which are the parameters considered to divide the boars in the group H and L.
3. We discover at the beginning of the results that there is a quality control group (QC) and we don't know if this group is either H, L or another group of animals.
4. Materials and methods is not a list of materials employed for doing the experiment, it should be the explanation of the experimental design, which in this paper is very confused. It is a pity because, maybe, results could have been interesting, but without these information it is not possible to fully understand them.
Also the form of the introduction is really below the minimum level of comprehension, it should be rewritten in a better English form, with the intent to let the reader fully understand not only the scope of the research experiment, but also what has already been stated in literature about the topic faced here.
Comments on the Quality of English Language
The quality of the English and especially the level of the details provided for the explanation of the background in the introduction and of the research design in Materials and Methods should be improved.
Author Response
Thanks very much for your professional advice!
Comments 1: Just at the beginning of the discussion, we discover the number of Boars included in the study. that should be clearly stated and moved in M&M, together with an explanation of how they have been divided in groups and how many ejaculates per each animal have been collected. It is not clear which are the parameters considered to divide the boars in the group H and L. |
Response 1:Thank you for pointing this out. I/We agree with this comment.” 2.2.4. grouping of boars and separation of SP” is added to explain these question. |
Comments 2: We discover at the beginning of the results that there is a quality control group (QC) and we don't know if this group is either H, L or another group of animals. |
Response 2: The QC was made by mixing all the samples, and three technical replicates were done. |
Comments 3: Materials and methods is not a list of materials employed for doing the experiment, it should be the explanation of the experimental design, which in this paper is very confused. It is a pity because, maybe, results could have been interesting, but without these information it is not possible to fully understand them. |
Response 3: First, boars and SP were divided into high and low freezability by progress motility. Then reproduction hormones, metabolites and proteins were examined to select matters related to freezablity. Then whether SP can be used to improve the sperm freezing effect is tested. SP can also be used in thawing solution, but that’s another parallel experimet, not related with freezability. |
Round 2
Reviewer 3 Report
Comments and Suggestions for Authors
I am sorry, but the reviewed paper, as revised in the present form, should be rejected in my opinion for some serious flaws who makes it to be not suitable enough for a journal like Animals.
First, the introduction is too poor to explain the complexity of the problem addressed by the authors. More insights are needed to the reader, and more references are needed in this section in order to give the reader a thorough and wider idea on the problem.
The part which is more problematic its the material and methods, where details also not useful for the clarity of the paper are mixed with some lacking information which are useful for the reader. example: no info on the extender composition are given, and even if a little more info on the animals used hase been added, still it was not clear about the research design. Why you put the ejaculates of the same animal in both bad or good freezer? Why you mixed the seminal plasma of three boars before the analysis? these are questions which are not answered in the text, but makes the research design really confused and not clear to some people who does clinical work.
To me, this work is too much oriented to biotechnolgies and too few to the clinical implications on these kind of research, to make it suitable for a journal like Animals.
Author Response
Comments 1: First, the introduction is too poor to explain the complexity of the problem addressed by the authors. More insights are needed to the reader, and more references are needed in this section in order to give the reader a thorough and wider idea on the problem. |
Response 1:Thank you for pointing this out. However the studies between freezability and seminal plasma are not too much. This can also be reflected in the discussion. The effects of many proteins and metabolites on freezing resistance are unknown. This reverse highlights the value of this study. We think that less relevant references are barely even less appropriate here. |
Comments 2: The part which is more problematic its the material and methods, where details also not useful for the clarity of the paper are mixed with some lacking information which are useful for the reader. example: no info on the extender composition are given, and even if a little more info on the animals used hase been added, still it was not clear about the research design. |
Response 2: The extender in the research was a commercial one, and the supplier has been given in the manuscript. In the manuscript , we has given the breed of boar : French Landrace or Yorkshire. In the pig farm, the boars usually are on (active) service from 10 month to 3 years old. So the age of the boars are also very limited. The design of the study is as clear as the order of results. Firstly, highly and lowly freezability boars and their SP ware selected, then the components of SP were tested by ELISA, metabonomic and protoemic methods to identity the DEMs and EDPs. Then bioinformatics method was used to analyze which pathways these DEPs and DEMs were enriched into. At last, the actual protective function of SP needs to be verified by using SP as base extender. |
Comments 3: Why you put the ejaculates of the same animal in both bad or good freezer? |
Response 3: Just to compare the protective effects of SP with high and low freezability. |
Comments 4:Why you mixed the seminal plasma of three boars before the analysis? |
Response 4: Mixing samples is a routine operation to reduce variation. When we only pay attention to the average value of the sample population, mixed samples can get more accurate estimates at less cost. |
Comments 5: this work is too much oriented to biotechnolgies and too few to the clinical implications on these kind of research. |
Response 5: Although the omics methods are used, the starting point and the end result of this study are how to improve the production effect of pig frozen semen. |